# *WRKY* Gene Family Drives Dormancy Release in Onion Bulbs

**DOI:** 10.3390/cells11071100

**Published:** 2022-03-24

**Authors:** Guglielmo Puccio, Antonino Crucitti, Antonio Tiberini, Antonio Mauceri, Anna Taglienti, Antonio Palumbo Piccionello, Francesco Carimi, Martijn van Kaauwen, Olga Scholten, Francesco Sunseri, Ben Vosman, Francesco Mercati

**Affiliations:** 1Dipartimento di Scienze Agrarie, Alimentari e Forestali, Università degli Studi di Palermo, Viale delle Scienze, 90128 Palermo, Italy; gugpuccio@gmail.com; 2Dipartimento Agraria, Università Mediterranea di Reggio Calabria, 89124 Reggio Calabria, Italy; antonino.crucitti@wur.nl (A.C.); antonio.mauceri87@gmail.com (A.M.); francesco.sunseri@unirc.it (F.S.); 3Plant Breeding, Wageningen University and Research Centre, P.O. Box 386, 6700 AJ Wageningen, The Netherlands; martijn.vankaauwen@wur.nl (M.v.K.); olga.scholten@wur.nl (O.S.); ben.vosman@wur.nl (B.V.); 4CREA Research Centre for Plant Protection and Certification, 00156 Roma, Italy; antonio.tiberini@crea.gov.it (A.T.); anna.taglienti@crea.gov.it (A.T.); 5Dipartimento di Scienze e Tecnologie Biologiche, Chimiche e Farmaceutiche-STEBICEF, Università degli Studi di Palermo, Viale delle Scienze Ed. 17, 90128 Palermo, Italy; antonio.palumbopiccionello@unipa.it; 6National Research Council of Italy, Institute of Biosciences and Bioresources (CNR-IBBR), 90128 Palermo, Italy; francesco.carimi@ibbr.cnr.it

**Keywords:** *Allium cepa* L., onion yellow dwarf virus, de novo transcriptome assembly, transcription factor, RNA-seq, biotic stress

## Abstract

Onion (*Allium cepa* L.) is an important bulb crop grown worldwide. Dormancy in bulbous plants is an important physiological state mainly regulated by a complex gene network that determines a stop of vegetative growth during unfavorable seasons. Limited knowledge on the molecular mechanisms that regulate dormancy in onion were available until now. Here, a comparison between uninfected and onion yellow dwarf virus (OYDV)-infected onion bulbs highlighted an altered dormancy in the virus-infected plants, causing several symptoms, such as leaf striping, growth reduction, early bulb sprouting and rooting, as well as a lower abscisic acid (ABA) level at the start of dormancy. Furthermore, by comparing three dormancy stages, almost five thousand four hundred (5390) differentially expressed genes (DEGs) were found in uninfected bulbs, while the number of DEGs was significantly reduced (1322) in OYDV-infected bulbs. Genes involved in cell wall modification, proteolysis, and hormone signaling, such as ABA, gibberellins (GAs), indole-3-acetic acid (IAA), and brassinosteroids (BRs), that have already been reported as key dormancy-related pathways, were the most enriched ones in the healthy plants. Interestingly, several transcription factors (TFs) were up-regulated in the uninfected bulbs, among them three genes belonging to the *WRKY* family, for the first time characterized in onion, were identified during dormancy release. The involvement of specific *WRKY* genes in breaking dormancy in onion was confirmed by GO enrichment and network analysis, highlighting a correlation between *AcWRKY32* and genes driving plant development, cell wall modification, and division via gibberellin and auxin homeostasis, two key processes in dormancy release. Overall, we present, for the first time, a detailed molecular analysis of the dormancy process, a description of the *WRKY-TF* family in onion, providing a better understanding of the role played by *AcWRKY32* in the bulb dormancy release. The TF co-expressed genes may represent targets for controlling the early sprouting in onion, laying the foundations for novel breeding programs to improve shelf life and reduce postharvest.

## 1. Introduction

Onion (*Allium cepa* L.) is one of the most cultivated vegetable crops worldwide (FAOSTAT, 2019. http://www.fao.org/faostat/en/#data (accessed on 12 January 2022). It is mainly used for human consumption as a vegetable and spice, raw or cooked, being a basic ingredient of all major diets [1,2]. Onion is a biennial crop that forms a bulb in the first year, which serves as a storage organ for the second year when flowering takes place. Between the first and the second year, the bulb undergoes a dormancy stage. Dormancy can be defined as a temporary suspension of visible growth of any plant structure containing a meristem aimed at surviving an unfavorable season [3]. In onion, dormancy occurs between bulb maturity and internal sprouting (Figure 1A). Several studies described the role and the crosstalk between different hormones in dormancy release, fructans mobilization, and external sprouting [4,5,6,7,8]. Abscisic acid (ABA), a growth hormone known to be involved in maturation and dormancy, accumulates during bulb growth [9]. After harvest, ABA is gradually depleted and reaches the lowest level just before sprout elongation [10]. By contrast, cytokinin (CK) and gibberellin (GA), correlated to sprout elongation, show opposite behaviors [8,11]. During the shift from dormancy to sprouting, the bulb converts from a sink to a source organ, allowing for cell division in the shoot apical meristem, directly on the basal plate. These changes involve cell wall modifications, hormones, and secondary metabolites synthesis [6]. The release of bulb dormancy is characterized by the mobilization of carbohydrate reserves as well as the cell elongation in sprout leaves, leading to a visible sprout from the neck of the bulb [12,13]. It is still unclear when dormancy in onion exactly ends [7]. Therefore, a better understanding of physiological and molecular mechanisms involved in dormancy and its breakage is crucial. Several factors have been reported to influence the release of dormancy, including genotype, harvesting time and technique, number of dry skins of the bulb, curing and storage conditions, carbohydrate metabolism, hormonal regulators, and pathogen infections [14,15,16,17]. The early dormancy release for instance due to virus infections is an important issue and may result in post-harvest losses.

The knowledge on dormancy regulation and release is still incomplete, and the role of transcription factors needs further investigation. Transcription factors (TFs) play a central role in these processes and can regulate gene expression through specific motifs usually found in the promoter gene region [18]. The *WRKY* gene family is an important and conserved class of TF and many efforts have focused on the identification of *WRKY* gene family members in several plant species [19,20,21,22,23,24,25,26]. Several members of the *WRKY TF* family are involved in biotic and abiotic stress responses [22,26,27,28,29]. Besides, *WRKY* TFs also play a significant role in plant development, including flowering, bud formation, and seed dormancy release [23,28,30,31,32,33]. Interestingly, the presence of differentially expressed *WRKY* genes at different stages of dormancy, appears to be conserved among various plant species from *Arabidopsis* to tree species. Moreover, a recent study identified a *WRKY* gene family member as a promoter of rice seed dormancy release via the ABA pathway [34]. In addition, a study in peony, a geophyte that forms a fleshy underground storage organ to overwinter, suggested that a *WRKY* gene might be involved in the early sprouting and in determining the chilling requirement for dormancy release [35]. In onion, the identification of the *WRKY* gene family members and their possible involvement in bulb dormancy remains to be elucidated.

In the present work, using de novo assembled transcriptome data and the onion genome reference sequence [36], we provide for the first time a comprehensive overview of the *WRKY* family members in onion. Our research evaluated their expression levels and the mechanisms induced at three different stages of dormancy. Comparative transcriptome analysis of onion yellow dwarf virus (OYVD)-infected and uninfected plants suggested a central role of *AcWRKY32* in onion bulb dormancy release.

## 2. Materials and Methods

### 2.1. Plant Material

Onion (*Allium cepa* L.) bulbs cv. “Rossa di Tropea” were grown in an experimental field trial in Campora San Giovanni (CS—Italy; 40°18′23.076″ N; 15°17′35.88″ E). Virus-free onion seeds were sown in January 2018 and transplanted in an insect-free greenhouse under natural conditions for temperature and light, applying water at 20 L m^−2^/plant in 12 h cycles, and providing fertilizer (one application of 100 g/plant of diammonium phosphate) management. A randomized block design was utilized and 180 plants were split into 6 experimental units (30 plants for each one). Artificial virus infection was carried out on 3 randomly selected plots, in two-month-old onion plants in March 2018 by mechanical inoculation using the potyvirus OYDV as described by Taglienti et al. [37]. Bulbs from each block (three biological replicates per each condition) were randomly collected at three different time-points: T_1_ (May 2018), just after harvest; T_2_ (July 2018), just before storage of the cured bulbs; and T_3_ (September 2018), just after the storage stage ended, and before the bulbs were planted. Onion plants were harvested when more than 80% of the leaves fell over (T_1_). Harvested plants were left on the soil to dry allowing the leaves to cover and protect the bulbs from sunlight. After bulb curing, when the green leaves were fully dried (T_2_), storage took place in a high-ceiling warehouse under dark conditions at room temperature (25 ± 0.5 °C), with uninfected and infected bulbs kept separately in plastic boxes. “Rossa di Tropea” is a cultivar not suitable for long storage and at the end of its usual storage stage (T_3_) by late September, is ready to be transplanted in autumn for seed production. Based on the physiological biennial onion life cycle (Figure 1A), the identified time points correspond to the beginning (T_1_), late (T_2_), and dormancy release (T_3_) stages. Visible morphological traits (e.g., external sprouting and rooting rates) were monitored at three time points and in both conditions (uninfected and infected samples).

### 2.2. OYDV Infection Evaluation Using DAS-ELISA and Real Time RT-PCR

The effectiveness of OYDV inoculation was verified as reported by Taglienti et al. [37], by an OYDV-specific serological test (ELISA) (BIOREBA, Reinach, Switzerland), following the manufacturer’s instructions. Inoculated plants negative at the ELISA test were further tested by quantitative real-time PCR (qPCR) to verify the infection efficiency [38]. Assays were performed in triplicate, including negative (uninfected plants and water) and positive (OYDV-infected plants by ELISA) controls.

### 2.3. Abscisic Acid (ABA) Quantification by Means of HPLC/MS

The ABA concentration was assessed by HPLC (Agilent 1260 Infinity) at each sampling time (T_1_, T_2_, and T_3_) using uninfected and OYDV-infected bulbs. Water and acetonitrile were of HPLC/MS grade. Formic acid was of analytical quality. Samples for HPLC were prepared by extracting ground material (100 mg) with MeOH (1 mL) for 24 h, at 4 °C. The supernatant was recovered after centrifugation (2000 rpm for 4 min) and used directly for injection using a reversed-phase C18 column (Phenomenex ZORBAX Extended-C18 2.1 × 50 mm, 1.8 µm), with a Phenomenex C18 security guard column (4 mm × 3 mm). The flow rate was set at 0.4 mL/min and the column temperature at 30 °C. The eluents were formic acid–water (0.1:99.9, *v*/*v*) (phase A) and formic acid–acetonitrile (0.1:99.9, *v*/*v*) (phase B). The following gradient was employed: 0–8 min, linear gradient from 5% to 50% B; 8–15 min, washing and reconditioning of the column to 5% B. Injection volume was 10 µL. The eluate was monitored through MS total ion chromatogram (TIC). Mass spectra were obtained on an Agilent 6540 UHD accurate-mass Q-TOF spectrometer equipped with a Dual AJS ESI source working in negative mode. N_2_ was employed as desolvation gas at 300 °C at a flow rate of 9 L/min. The nebulizer was set to 45 psig. The sheath gas temperature was set at 350 °C at a flow rate of 12 L/min. A potential of 2.6 kV was used on the capillary for negative ion mode. The fragment was set to 75 V. MS spectra were recorded in the 100–1000 *m*/*z* range. The ABA standard and other chemicals were supplied by Sigma-Aldrich (St. Louis, MO, USA) and used as received. The calibration curve was performed in a 0.01–10 µg/mL range (R^2^ = 0.9995), and the results were corrected for the weight of the plant sample and expressed as µg of ABA per gram of onion.

### 2.4. RNA Isolation and Sequencing

Total RNA was isolated using a NucleoSpin RNA Plant (Macherey-Nagel GmbH & Co., KG, 52355 Düren, Germany) minikit for RNA extraction and treated with RNase-free DNase. Three biological replicates for each of the time points and conditions were used. RNA quality (RNA integrity number—RIN > 8.0) was assessed using an Agilent Bioanalyzer RNA nanochip (Agilent, Wilmington, DE, USA). Sequence libraries were prepared using a TruSeq RNA Sample Preparation Kit v2 (Illumina, San Diego, CA, USA), according to the manufacturer’s instructions. Both quality and insert size distribution were assessed using an Agilent Bioanalyzer DNA 1000 chip. Sequence libraries were quantified using qPCR (P5/P7 primers), pooled in equimolar concentration, and analyzed on an Illumina NextSeq500 generating 2 × 150 nt paired-end (PE) reads. The generated sequences were deposited in the NCBI (National Center for Biotechnology Information) PRJNA797248.

### 2.5. De Novo Assembly, Annotation, and Differential Gene Expression Analysis

Raw reads from each sample were assessed for quality using the FastQC tool v0.11.8 (https://www.bioinformatics.babraham.ac.uk/projects/fastqc/ (accessed on 10 November 2021). Adapters were removed and the quality trimming was performed using Trimmomatic v0.38 (http://www.usadellab.org/cms/?page=trimmomatic (accessed on 10 November 2021). Reads were filtered by length and only those longer than 36 bp were selected. The filtered reads were pooled into a single dataset and a de novo transcriptome was assembled using Trinity v2.8.4 (https://github.com/trinityrnaseq/trinityrnaseq/releases (accessed on 10 November 2021). The Cd-hit (v4.7) tool (http://weizhongli-lab.org/cd-hit/ (accessed on 17 November 2021) was used to reduce the number of transcripts and to collapse highly similar sequences, with default parameters. Assembly completeness was assessed with BUSCO v2/v3, using the gVolante web service (https://gvolante.riken.jp/ (accessed on 17 November 2021), and the Mercator online tool v. 4.0 (https://mapman.gabipd.org/app/mercator (accessed on 17 November 2021) was utilized for functional annotation.

Transcript quantification was performed using Salmon (https://salmon.readthe-docs.io/en/latest/salmon.html (accessed on 17 November 2021). All the sampling times (T_1_, T_2_, and T_3_) within each treatment and between uninfected and OYDV-infected bulbs at each time were compared to identify the differentially expressed genes (DEGs) (Padj < 0.05). DEGs identification and principal component analysis (PCA) were carried out by utilizing Deseq2 (http://www.bioconductor.org/packages/release/bioc/html/DESeq2.html (accessed on 17 November 2021). The MapMan tool (http://gabi.rzpd.de/projects/MapMan/ (accessed on 17 November 2021) was utilized to assign specific metabolic pathways to extracted DEGs. The R packages VennDiagram (https://cran.rproject.org/web/packages/VennDiagram/index.html (accessed on 17 November 2021) and topGO (http://bioconductor.org/packages/release/bioc/html/topGO.html (accessed on 17 November 2021) were used to draw Venn diagrams and for Gene Ontology (GO) enrichment analysis, respectively.

### 2.6. WRKY Extraction and Validation in the Onion Genome Sequence

The NCBI ORFfinder was used to predict and translate the putative ORFs for each transcript (https://www.ncbi.nlm.nih.gov/orffinder/ (accessed on 17 November 2021). To search the ORF dataset with the Pfam profile of the WRKY protein family (PF03106), HMMER hmmscan v3.2.1 (http://hmmer.org/ (accessed on 17 November 2021) was used. All expressed *AcWRKYs* were extracted for each sampling time and under both conditions, identifying the genes belonging to the *WRKY* family that were missing from the Mercator annotation. After selecting the best-scoring isoform from the hmmscan results, sequences were filtered for the presence of the complete WRKY domain and sequence length (>40 AA) with a custom bash script. Transcripts were then annotated with a blastp v2.7.1+ search (https://kbase.us/applist/apps/kb_blast/BLASTp_Search (accessed on 17 November 2021) in the NCBI nr database and aligned with the Clustal Omega tool (http://www.clustal.org/omega/ (accessed on 17 November 2021). Finally, the *WRKY* TFs isolated were validated by mapping their sequences to the onion genome recently assembled [36].

### 2.7. Phylogenetic Analysis and Structural Organization of AcWRKY

*WRKY* genes from five different taxa (*Solanum lycopersicum*, *Arabidopsis thaliana*, *Triticum aestivum*, *Oryza sativa* ssp. *Japonica,* and *Vitis vinifera*) were downloaded from the Plant Transcription Factor Database (PlantTFDB; http://planttfdb.gao-lab.org/ (accessed on 24 November 2021). The final dataset used for the phylogenetic analysis included 587 protein sequences summarized in Table 1. Alignment was performed with the online tool CLUSTALW with default parameter settings. The unrooted phylogenetic tree was generated through Iqtree software v. 2-1-3 (http://www.iqtree.org/ (accessed on 24 November 2021) with the maximum likelihood method, 1000 bootstrap replicates, and the JTT + G4 model (Iqtree best-fit model selection), and visualized by FigTree v. 1.4.4 (http://tree.bio.ed.ac.uk/software/figtree/ (accessed on 24 November 2021) and iTol v.6 (https://itol.embl.de/ (accessed on 24 November 2021). A dendrogram based on the multiple sequence alignment of onion *WRKY* genes (*AcWRKY*) was developed using the same parameters and the VT + F+G4 model (Iqtree best-fit model selection). Finally, to discover conserved motifs among *AcWRKYs* and to investigate the WRKY domains and their organization, the MEME online tool v5.0.5 (http://meme-suite.org/ (accessed on 24 November 2021) was utilized.

### 2.8. Validation of Differentially Expressed AcWRKY Genes by RT-PCR

Quantitative real-time PCR (qPCR) was used to validate the differentially expressed *AcWRKYs* isolated in the comparisons and putatively involved in onion dormancy modulation. The *AcWRKY* sequences, previously validated on the onion genome [36], were used to design primer pairs for each candidate TF using Primer 3.0 software (http://primer3.ut.ee/ (accessed on 24 November 2021), which can be found in Appendix A. Reverse transcription was performed on 200 ng of total RNA extracted from uninfected and OYDV-infected bulbs at T_1_, T_2_, and T_3_, using iScript Reverse Transcription Supermix (Bio-Rad, Berkeley, CA, USA), according to manufacturer’s instructions. qPCR was performed using 1.5 μL cDNA diluted 1:10, 0.3 μM forward and reverse primers and 5 μL of iQ SYBR Green Supermix (Bio-Rad, Berkeley, CA, USA) in 10 μL final volume, with the following thermal program: initial activation at 95 °C for 3 min, 40 cycles at 95 °C for 15 s and annealing temperature (60 °C) for 1 min. This procedure was followed by melting curve analysis from 95 °C for 10 s, 65 °C for 5 s, to 95 °C for 5 s. The actin gene was used as reference [39] and each sample was analyzed on three biological and two technical replicates. Fragment amplification was verified by 1.5% *w/v* agarose gel electrophoresis and melting curve analysis; PCR efficiency of primer pairs was optimized to be in the range 92–100% with R^2^ values of 0.996. The relative expression ratio of each gene was calculated by the 2^−ΔCT^ method [40]. Pearson correlation analysis between RNA-Seq and qPCR was performed.

### 2.9. Networking Analysis

A co-expression network analysis was performed using the Comparative Co-Expression Network Construction and Visualization tool (CoExpNetViz) (http://bioinformatics.psb.ugent.be/webtools/coexpr/ (accessed on 2 December 2021). Normalized expression data extracted from Deseq2 were used to calculate Pearson’s correlation coefficient using the 1st and 99th percentiles of the correlation distribution as thresholds. Differentially expressed *AcWRKYs* across sampling times and between OYVD-infected and uninfected conditions were used as baits. The resulting network was visualized by Cytoscape v.3.9.1 (https://cytoscape.org/ (accessed on 2 December 2021).

## 3. Results

### 3.1. Infection of Onion with OYDV

The inoculation efficiency was assessed by both ELISA and qPCR. The ELISA assay showed an evident and significant reaction in both the positive controls and 87% of the inoculated plants; signal was not detected in both negative controls and uninfected samples. Inoculated plants without signal in the ELISA reaction were also investigated by qPCR. Positive controls showed Ct values ranging from 16 to 18, while inoculated bulb samples, negative at the ELISA test, showed Ct values from 18 to 21, confirming their OYDV correct artificial infection. Uninfected plants and water (negative control) did not show amplification, confirming both the salubrity of the uninfected samples and the assay specificity.

### 3.2. ABA Concentration and Morphological Changes in Uninfected and OYDV-Infected Bulbs during Different Stages in Bulb Dormancy

At the beginning of dormancy (T_1_), higher ABA levels in uninfected compared to OYDV-infected bulbs (*p* < 0.001) were observed (Figure 1B; Appendix A). In the uninfected samples, ABA decreased from T_1_ to T_3_, whereas the opposite trend was observed in the OYDV-infected bulbs, showing the highest hormone concentration at T_3_ (Figure 1B; Appendix A). At T_3_, sprouting and rooting were observed in the OYDV-infected bulbs, while external sprouting was not visible in the uninfected ones. (Figure 1C; Appendix A).

### 3.3. Transcriptomic Profiles of Infected and Uninfected Onion Bulbs Comparing the Dormancy Stages at Three Time Points

Transcriptome analysis of infected and uninfected onion bulbs was carried out at three dormancy stages (Figure 1A). Sequencing statistics are presented in Appendix A. The comparisons between the dormancy stages (T_2_ vs. T_1_; T_3_ vs. T_1_, and T_3_ vs. T_2_) within each treatment resulted in 5390 and 1322 DEGs in the uninfected and the OYVD-infected samples, respectively (Appendix A), showing a lower number of up- and down-regulated genes in the latter condition (Appendix A). In the uninfected bulbs, the highest number of DEGs in the T_3_/T_1_ comparison (2278) was found, while the OYVD-infected samples displayed the highest DEGs number in the T_3_/T_2_ comparison (617). The lowest number of DEGs was in the T_2_/T_1_ comparison in both conditions (Appendix A). By contrast, in both comparisons, T_3_/T_1_ and T_3_/T_2_, the number of up-regulated was higher than the down-regulated genes, but only in the uninfected plants (Appendix A). Cell wall modification, proteolysis, hormone signaling, transcription factors, protein degradation, receptor-like kinase, as well as nitrate transporters, and amino acid biosynthesis were the most prevalent categories at dormancy release (T_3_) in the uninfected bulbs (Appendix A). Finally, many transcripts related to ethylene signal transduction, auxin-responsive family (such as SAUR-like protein), and several TFs belonging to WRKY, MYB, and MAPKs were higher at T_3_ compared to both T_2_ and T_1_ (Appendix A).

GO terms enrichment analysis highlighted clear differences between uninfected and OYVD-infected bulb profiles at each sampling time (Appendix A). In the biological process (BP) category, GO terms “cell wall organization” and “cell wall biogenesis” were found among the up-regulated genes in the uninfected samples at T_2_ and T_3_, together with “carbohydrate metabolic process” and “regulation of jasmonic acid-mediated signaling pathway”. In addition, at T_3_ in the same category, “regulation of transcription”, “response to chitin”, “response to cold”, “response to jasmonic acid “, and “signal transduction” categories were identified up-regulated only in uninfected bulbs (Appendix A). By contrast, OYDV-infected plants showed a very different pattern with only few BP GO terms enriched at each time point (Appendix A). Accordingly, many of the enriched GO terms in the molecular function (MF) category were only detected at T_3_ in uninfected profiles, including “DNA-binding transcription factor activity”, “carbohydrate binding”, “polysaccharide binding”, “calmodulin binding”, and “xyloglucan: xyloglucosyl transferase activity (Appendix A). Finally, in the cellular component (CC) category, only a few categories were enriched, mainly related to uninfected bulbs, such as “plant-type cell wall”, “extracellular region”, “apoplast”, and “anchored component of plasma membrane” (Appendix A).

### 3.4. Differentially Expressed Genes between OYDV-Infected and Uninfected Bulbs at Each Dormancy Stage

To identify genes linked to dormancy and select candidate genes involved in its release, the transcriptome profiles of uninfected and OYDV-infected onion bulbs across the three time points were compared (Appendix A). Overall, the OYDV-infected bulbs showed a low and stable DEGs number during the three time points (Appendix A). PCA analysis distinguished uninfected and OYVD-infected samples, with a clearer separation among the three dormancy stages in the uninfected bulbs (Appendix A). Thus, the expression profiles of genes involved in several pathways related to key physiological processes, such as cell wall modification, transcription factors, receptor-like kinases, and hormone signaling, showed significant differences between the infection conditions (Appendix A). In the ABA—dependent signaling pathway, the 9-cis-epoxycarotenoid dioxygenase (*NCED3*) and the abscisic acid 8′-hydroxylase 1 (*CYP707A1*) as well as the *ABI5* leucine zipper transcriptional factor (ABA Insensitive 5)*,* the pre-mRNA polyadenylation factor (*FIP1*), the ABA-hypersensitive germination 3 (*AHG3*), and the GL2-expression modulator (*GEM*) were differentially expressed between bulb infection conditions. In detail, *NCED3* and *CYP707A1*, involved in the ABA accumulation [41], were down-regulated (log2 FC = −3.231; log2 FC = −4.459) in OYDV-infected bulbs at T_2_ and T_3_, respectively. By contrast, *GEM*, and the polyadenylation factor *FIP1*, expressed in the meristems during cell division [42], were up-regulated (log2 FC = 6.035 and log2 FC = 1.206, respectively) in uninfected bulbs at T_3_, as well as *AHG3* (log2 FC = 3.207) involved in seed dormancy release [43]. In agreement, at dormancy breakage (T_3_), *ABI5* was down- (log2 FC = −3.053) and up-regulated (log2 FC = 1.243) in uninfected and OYDV-infected bulbs, respectively. Seven genes encoding for SAUR-like auxin-responsive family proteins as well as a nitrate transporter (*NRT1.1*; log2 FC = 3.306) were up-regulated in the uninfected bulbs at T_3_. The *IAA4*, a short-lived TF working as a repressor of early auxin response genes [44], and the auxin efflux carrier *PIN7* were also up-regulated in the uninfected bulbs at T_2_ (log2 FC = 1.3091) and T_3_ (log2 FC = 4.0866), respectively (Figure 2; Appendix A). Furthermore, three gibberellin (GA) related genes, the Ent-kaurenoic acid oxidase 2 (*KAO2*), the gibberellin 3-beta-dioxygenase 1 (*GA3OX1*), and the gibberellic acid requiring 1 (*GA1*), as well as the NAC1 transcription factor were up-regulated in the uninfected compared to OYVD-infected samples at dormancy release (T_3_). Remarkably, using the MapMan annotation, eight genes belonging to the *WRKY* family, an essential class of transcriptional regulators involved in key biological processes and linked to the pathways listed before [23,28], resulted differentially expressed in the uninfected bulbs at T_3_ (Figure 2; Appendix A). Since the genes belonging to this TF class might be considered potential candidates involved in dormancy modulation, as recently described in rice [34], they were further explored for their putative role in onion dormancy release.

### 3.5. Characterization of WRKY Genes in the Onion Genome

To characterize the *WRKY* family members in onion, NCBI’s ORFfinder was used. Fifty-two (52) open reading frames (ORFs), present in all the sequence data (T_1_–T_3_), were predicted to belong to *AcWRKYs* in the de novo transcriptome assembly. These sequences were mapped and validated in the onion reference genome [36]: forty-six out of 52 showed a sequence identity >90% (Appendix A). Six *WRKY* transcripts (*AcWRKY22, 24*, *25*, *35*, *49*, and *51*) were discarded from the analysis either because of their short amino acid sequence (<40 aa) or because of the lack of a strong hit in the reference genome. Finally, a genome-wide BLAST search for protein sequences containing a WRKY domain in the reference genome identified 28 additional WRKY genes (Appendix A). Overall, the *AcWRKY* panel included 74 genes of which 16 with two WRKY domains were assigned to group I; 32 genes were included in group II (C2H2 domain) and the remaining 25 belonged to group III (C2HC domain). Only *AcWRKY*16 did not group in any of the three groups (Appendix A).

### 3.6. Phylogenetic Analysis of Onion WRKY Genes

The relationships among the *AcWRKY* genes were investigated using protein sequence similarity. Two maximum likelihood trees were constructed: (i) a first based on a wide dataset including 587 full-length genes encoding WRKY proteins belonging to six plant species, including *Allium cepa* (Table 1; Figure 3), and (ii) the second one including only 74 *AcWRKY* sequences isolated here (Appendix A). In the first, clusters among groups and subgroups were evident, regardless of the species. *AcWRKY*s clustered in three main groups in agreement with the previous *WRKY* family classification [45].

Genes belonging to group II were distinguished in five subgroups (IIa, IIb, IIc, IId, and IIe), based on their primary amino-acid sequences. All subgroups, as well as group III, were monophyletic except groups IIb and IId (Figure 3). Twelve previously unclassified *WRKY* genes from different species [21,22,23,24,25,26], including the *AcWRKY16* gene, clustered (highlighted in yellow) and belonging to subgroup IIc (Table 1; Figure 3). The consensus phylogenetic tree obtained by aligning only the AcWRKY protein sequences produced five main clusters (Appendix A). Group III (C2HC domain) showed higher variability than other groups forming multiple subclusters. Instead, groups I and II and the five subgroups of group II were distinguished in clusters with single membership. Groups IId and IIe clustered, while group I was close to groups IIa, IIb, and IIc. Four *AcWRKY* genes belonging to group IIc (*AcWRKY14* and *AcWRKY73*) and IId (*AcWRKY5* and *AcWRKY65*) clustered to group I. Finally, *AcWRKY16* did not fall into any of the main *WRKY* groups (Appendix A).

The structural organization of *AcWRKYs* was also analyzed through the MEME tool, showing the conserved motifs for each gene (Appendix A). *AcWRKY* groups and subgroups shared specific motif patterns [46], in agreement to the phylogenetic tree (Appendix A). Almost all WRKY domains in group I and group II were characterized by three conserved motifs, Motif1, Motif3, and Motif2, which contain the WRKYGQK and the zinc-finger-like domains, while the second WRKY domain of group I, in the C-terminus region, was preceded by Motif5. Finally, group III showed the most variable structural organization with subgroups characterized by WRKY domains without zinc-finger motifs and the Motif4 + Motif1 WRKY organization. Group-specific motifs were found in all three groups. In particular, Motif4, Motif7, Motif15, and Motif20 were exclusive to group III, Motif10 and 19 to groups IIb and IIe, respectively, while Motif13 only belonged to group I.

### 3.7. AcWRKYs Differentially Expressed during Dormancy Different Stages

Fourteen differentially expressed *AcWRKYs* (#1, 2, 3, 9, 12, 17, 18, 21, 26, 30, 32, 37, 46, and 52) in the uninfected compared to the virus-infected bulbs at different time samplings were found (Appendix A). By contrast, *AcWRKY* genes were not differentially expressed in OYDV-infected bulbs, comparing all the time samplings (Appendix A). Three out of the fourteen differential expressed *AcWRKYs* (*AcWRKY2*, *AcWRKY30*, and *AcWRKY32*) showed a significantly higher expression level (*p* < 0.05; log2 FC > 1) in the uninfected compared to OYDV-infected bulbs at T_3_ (Figure 4). Transcriptome profiles of all differentially expressed *AcWRKYs* were validated by qRT-PCR and their expression trend in qRT-PCR was similar to the transcriptome expression pattern (Appendix A). A Pearson’s correlation analysis between RNA-seq and qPCR results showed that the correlation was 0.88 (R) and highly significant (*p* < 0.001) (Appendix A).

### 3.8. Gene Co-Expression Network Analysis

A co-expression network analysis was carried out to gain insight into the role of *AcWRKY*s play in dormancy breakage. Significantly up-regulated *AcWRKY* genes during dormancy release (T_3_) between uninfected and OYDV-infected bulbs (*AcWRKY2, 30, 32*), were used as baits to identify co-expressed genes (Figure 5). Three hundred and sixty-two (362) genes were co-expressed with the selected *AcWRKY*s (Figure 5; Appendix A). *AcWRKY32* and *AcWRKY2* were co-expressed with 180 (50%) and 169 (47%) genes, respectively, whereas only 13 genes were found in the network of *AcWRKY30*. In agreement with the MapMan and GO enrichment analyses, several genes encoding for MAP kinases, calcium-dependent protein kinases (CDPK), ethylene, brassinosteroids (BRs), ABA, jasmonic acid, and auxin-related proteins, together with TFs and cell cycle regulation proteins, were found to be co-expressed, mainly in the *AcWRKY32* network (Appendix A). In detail, the main auxin-carriers, *PIN7* (TRINITY_DN1680_c0_g1), the dual affinity *Nrt1.1* (TRINITY_DN4953_c0_g1) together with the high affinity nitrate transporters *Nrt*2.5 (TRINITY_DN14312_c0_g2) and the auxin response factor *ARF2A* (TRINITY_DN33314_c0_g1) were co-expressed with *AcWRKY32*. The somatic embryogenesis receptor kinase 1 (*SERK1*; TRINITY_DN35324_c0_g2), and the serine/threonine protein kinase *ARK3* (TRINITY_DN7704_c0_g1) also belonged to the *AcWRKY32* network.

In the same gene network, transcripts involved in calcium signaling, such as the plasma membrane calcium pump *ACA9* (TRINITY_DN1881_c0_g1), the calcium influx regulator *GLR2.8* (TRINITY_DN28372_c0_g1), the calmodulin binding protein *CML42* (TRINITY_DN23819_c0_g1), as well as the inositol polyphosphate 5-phosphatase *5PT* (TRINITY_DN5753_c0_g1), and genes related to cell wall development and biosynthesis (as *WALK20*—TRINITY_DN27969_c0_g2; *WAV3*—TRINITY_DN322_c3_g1; and *MWL-1*—TRINITY_DN6355_c0_g1) were included (Appendix A).

## 4. Discussion

The quality of onion bulb and its bio-compounds content are affected by several variables during harvesting and storage, which mainly relate to the dormancy duration, water content, and plant tolerance to pathogens [6,17]. The bulb dormancy is considered a critical step in the onion lifecycle, and one of the main factors for yielding bulbs of high quality and nutraceutical value, thus the increase of our understanding on the molecular mechanisms involved in dormancy and its breakage is crucial. In particular, those related to the early sprouting (dormancy release) are critical for crop management, mainly in moist environments [6,13]. Recently, the ubiquitous *WRKY* genes were proposed as a versatile toolbox involved in the modulation of numerous biological processes, such as plant development, biotic and abiotic stress responses, embryogenesis, leaf senescence, and seed and bud dormancy [22,23,26,28,31,47,48,49]. *WRKY* gene family analysis through NGS technologies was carried out in a high number of plant species [22,25,26,50], however, these TFs have not been characterized in onion, until now.

Here, a comparative analysis between uninfected and OYDV-infected bulbs across three main stages of dormancy was carried out to identify genes, regulatory factors, and molecular pathways involved in dormancy modulation and release, uncovering specific *AcWRKYs* which appear to be involved in the dormancy breakage (Appendix A).

### 4.1. Transcriptome and ABA Profiles during Onion Bulb Dormancy and Its Alteration Caused by OYDV Infection

The onion bulb has evolved as a storage organ that allows plants to over season by entering into the dormant state that enables the bulb to undergo the required morphological and molecular transformations to survive at adverse environmental conditions. Bulb dormancy ends with the beginning of the internal sprouting; however, the onion bulb commercial storage life continues until root and sprout elongation [17]. During the shift from dormancy to sprouting, the bulb undergoes a conversion from sink to source organ, allowing cell division in the basal plate. The changes involve cell wall modification, hormones, and secondary metabolites synthesis [6]. Pathogen infection is one of the main causes that can alter dormancy. Potato tubers infected with bacteria or fungi showed a reduced storage period with more visible sprouting than the healthy tubers [51]. Another example was reported in *Euphorbia pulcherrima*, in which virus-infected buds caused rooting of stem cuttings during dormancy while the healthy buds remained dormant [52]. Virus infection caused the growth of axillary shoots in tobacco [53] during storage while it shortened shelf-life due to early sprouting in *Allium* spp. [14]. We compared transcriptome profiles in OYVD-infected and uninfected bulbs to identify genes, regulator factors, and molecular pathways driving the dormancy process and its release in onion.

Almost five thousand four hundred (5390) DEGs were identified in uninfected samples comparing the dormancy stages (T_2_ vs. T_1_; T_3_ vs. T_1_, and T_3_ vs. T_2_) (Figure 1). Many of these expression changes involved genes related to primary and secondary metabolism, cell wall modification, and hormone signaling. In the OYVD-infected bulbs, only 1322 genes were differentially expressed in the same comparisons, which is consistently lower than the uninfected bulbs. In particular, the comparison T_3_/T_1_ identified 1129 genes significantly upregulated (Log2 fold change > 1) in the uninfected samples, compared to only 191 genes in the OYVD-infected bulbs at the same comparison. These findings confirmed the ability of a *Potyvirus*, such as OYVD, to induce host gene down-regulation in the infected onion bulbs/plants, as previously described in tomato and other plants [54,55].

Several genes involved in cell wall modification and cellular division-induced in uninfected samples at dormancy release (T_3_), were also found up-regulated during bud dormancy release in many plant species [56,57]. Among these, pectinesterase, pectate lyase, and chitinases are involved in the bulb softening through cell wall degradation. A recoding of transcripts related to cell cycle regulation is required to drive the storage compounds transport, providing new tissues needed for the incipient sprouting [6]. At the same stage (T_3_), GO terms related to sterols, hydrogen peroxide, calcium, and calmodulin were enriched in uninfected compared to OYVD-infected samples. During dormancy, sterols play an important role in plasma membrane permeabilization, promoting metabolites remobilization [58], while hydrogen peroxide was reported as a key element for seed dormancy release [59]. The transcriptome profiles highlighted a marked differential expression for several TF gene families, such as *ERF*, *bZIP*, *MYB*, *DOF*, and *WRKY*, between uninfected and OYDV-infected bulbs. Moreover, we detected a significant up-regulation of the ABA responsive gene *GEM* (GL2 expression modulator), found involved in cell division and germination regulation, during seed dormancy breakage in *Arabidopsis* [60], only in the uninfected bulbs at dormancy release (T_3_). Interestingly, ABA-hypersensitive germination 3 (*AHG3*), encoding for a PP2C phosphatase, was found up-regulated in the uninfected samples at T_3_, suggesting its role in bulb dormancy release. *AHG1,* a member of the same gene family, interacts with delay of germination 1 (*DOG1*) and ABA, positively regulating seed dormancy release [43,61].

These observations are consistent with the ABA levels measured in our samples from T_1_ to T_3_. Indeed, ABA levels are higher at the dormancy onset (T_1_) in uninfected than OYVD-infected bulbs, with quickly decreased levels at T_2_ and T_3_. The lower ABA levels in the infected bulbs at T_1_ suggested that dormancy did not yet take place. At T_3_, ABA levels increased only in infected samples, showing an opposite ABA profiling in the uninfected bulbs. Plant growth hormones seemed to play a crucial role in maintaining bulb dormancy or its breakage, determining sprout elongation [5,8,9,10,17]. In particular, ABA plays a central role in all plants in biological processes related to senescence and dormancy regulation. In onion, ABA was reported to be correlated to the repression of growth and to bulb dormancy at harvest time [9,62]. However, ABA plays different roles in many biological processes, depending mainly on the biological stage and status of the plant [63]. Several studies associated this hormone with the response to abiotic stresses [64]. At T_3_, the infected bulbs appeared already sprouted and ready for the next onion life cycle.

### 4.2. Onion WRKY Gene Characterization

In the present work, for the first time, 74 *AcWRKY* genes were identified in onion. The reference genome was defined on a doubled haploid line that differs from ”Rossa di Tropea”, thus a de novo assembly approach and annotation of the onion bulb transcriptome of our cultivar was used. Fifty-two (52) *AcWRKY* genes were de novo identified and validated, of which six were considered not reliable because of the short amino acidic sequence and the low percent identity. A genome-wide search in the reference genome yielded an additional 28 genes, unidentified in the transcriptome of “Rossa di Tropea”, probably due to tissue-specific expression, different timing of expression during the onion life cycle, or a stress-specific induction. The total number of *WRKYs* identified in onion is comparable to those found in *Arabidopsis*, grapevine, and tea plant [19,22,26]. Otherwise, the number of *WRKYs* is variable among plants and a higher number (from 100 to 174) was identified in other species, such as poplar, maize, and soybean [20,25,48]. Based on WRKY domain number and structure, the family members identified in onion can be divided into the main groups I, II, and III, previously defined [45]. A phylogenetic analysis that included *WRKY* genes from six different plants, including onion, agreed with the evolutionary lineage of *WRKY-TF* in flowering plants proposed by Rinerson et al. [65], highlighting four major clades, represented by groups I + IIc, groups IIa + IIb, groups IId + IIe, and group III, respectively. Interestingly, the few *WRKYs* belonging to different species, previously unclassified [26,48], clustered in sub-group IIc (Figure 3, yellow). These genes, including *AcWRKY16,* were characterized by a slightly different zinc-finger-like motif or WRKY domain. The phylogenetic tree of *AcWRKY* genes showed the same three groups proposed by Eulgem et al. [45]. Four *AcWRKY* genes belonging to group IIc (*AcWRKY14* and *AcWRKY73*) and IId (*AcWRKY5* and *AcWRKY65*) clustered in group I, probably due to their incomplete WRKY domains. Seventy-seven percent of *AcWRKY* genes showed at least one highly conserved heptapeptide motif WRKYGQK whereas in thirteen genes six variants of the heptapeptide were found: WRKYGEK (4), WRKYGKK (3), WRKYEQK (1), WKKYGEK (2), WHKYGEK (2), and WKKYGKK (1). These variations may be involved in the recognition of slightly different target sequences [18] or DNA binding efficiency [27]. Four *AcWRKY*s, lacking the heptapeptide motif, were found only in the reference genome, belonging to groups I (*AcWRKYY71)* and IIe (*AcWRKY69, AcWRKY75, AcWRKY80*). The lack of the heptapeptide motif could be related to a specific evolutionary deletion. Moreover, the *AcWRKY69* and *AcWRKY75* amino acid sequence and conserved motifs suggested a gene duplication event.

### 4.3. WRKY Potential Involvement in Onion Bulb Dormancy Release

The uninfected bulbs showed prolonged dormancy, and were used in comparison to infected bulbs to highlight genes, regulatory factors, and molecular pathways involved in dormancy modulation and breakage. Among them, genes belonging to the *WRKY* gene family seemed related to the dormancy release. The expression level of fourteen members of our *AcWRKY* panel was significantly higher in the uninfected compared to OYVD-infected bulbs over time sampling (from T_1_ to T_3_). At T_3_, three out of 14 genes (*AcWRKY2*, *AcWRKY30*, *AcWRKY32*) were up-regulated during dormancy release and showed higher expression in uninfected bulbs that were sprouted compared to OYDV-infected bulbs. *Arabidopsis* orthologous genes were reported to drive important physiological processes. *AtWRKY41* (=*AcWRKY2*) is involved in seed dormancy regulation [31]; *AtWRKY22* (=*AcWRKY30*) regulated the dark-induction of leaves’ senescence [66]; while, *AtWRKY6* (=*AcWRKY32*) was found to control senescence, seed germination, and early seedling development [67]. Interestingly, concurrent differential expression between three identified *AcWRKY*s and key regulator genes of plant developmental switches was found. The main plant development and growth-related genes were co-expressed with *AcWRKY32*. The co-expression network of *AcWRKY30* was comprised of 13 genes, providing limited information on its possible role, while the *AcWRKY2* network included 169 co-expressed genes, albeit they did not appear directly involved in pathways related to dormancy release. By contrast, genes from plant growth regulator pathways, such as auxin, ABA, ETH, and BR, as well as genes involved in their signaling pathways, differentially expressed during bud dormancy breakage in grape [68], were co-expressed with *AcWRKY32*, suggesting this TF as a candidate gene for the regulation of onion bulb dormancy release. Indeed, the *AcWRKY32* co-expression network included genes already related to dormancy release in other plants (Appendix A; Figure 5 and Figure 6). In particular, several small auxins up RNA (*SAUR*) genes, belonging to primary *AUXIN RESPONSE FACTORS* (*ARFs*), such as *ARF2A*, and putative auxin transporters (*PIN7*, *NRT1.1*, *NRT2.5*) were up-regulated only in the uninfected bulbs during dormancy release (T_3_). These expression levels agreed with previous studies, showing an increased expression of *SAUR* genes before bud breakage in grapevine [22]. Auxin and its influx carriers are required to maintain PM H^+^-ATPase activity, promoting cell expansion and elongation, two key events during dormancy release [69]. The IAA key role in cell expansion and division, driving the intensity of mitosis as well as the structural and functional changes occurring in cambial cells during dormancy, has been previously reported in woody plants [70]. Calcium (Ca^2+^) signaling genes (*ACA9*; *GLR2.8*), as well as calmodulin binding proteins (*CaBP*-*CML42*), involved in Ca^2+^ signal transduction, were also identified in the *AcWRKY32* co-expression network. Calcium signaling and calmodulin binding proteins have been previously associated with dormancy regulation and dormancy breakage in grape buds and *Arabidopsis* [71,72]. Moreover, Ca^2+^ and its signaling pathway are essential in both seed and pollen germination and seedling establishment, driving the early cellular events occurring during and after dormancy breakage [71,73]. Many *AcWRKY32* co-expressed genes, are involved in cell wall development and modification, including genes related to cell wall biosynthesis such as *WALK20* and *WAV3* (Figure 5 and Figure 6; Appendix A). Furthermore, the expression profile of these genes could be related to Ca^2+^ influx in the cytosol caused by GAs, the main group of phytohormones driving dormancy breakage [74]. Interestingly, genes involved in the gibberellin pathway, such as *GA1*, *GA3ox1*, and *KA02*, were up-regulated during dormancy release (T_3_) only in the uninfected bulbs, as expected. In agreement with our findings, *GA3ox1* and *GA1* up-regulation, both involved in the biosynthesis ofGA_1_, one of the main bioactive GAs, was detected during dormancy release in grape and *Arabidopsis* [75,76]. Furthermore, an increased *GA3ox1* transcript abundance and its interaction with *KAO2*, induced seed germination by breaking ABA inhibition in *Polygonatum kingianum* [77]. These findings are consistent with ABA levels reduction in uninfected samples between T_1_ and T_3_. A major gene involved in the ABA regulation and signaling, *ABA INSENSITIVE 5* (*ABI5)*, which encodes a member of the basic leucine zipper transcription factor family, was down-regulated at T_3_ in uninfected bulbs. A down-regulation of ABI5 that promotes both seed germination and seedling growth was reported in *Arabidopsis* [78], whereas its up-regulation in the vegetative stage is related to abiotic stress due mainly to water loss and osmotic stress [79]. Gene expression in the ABA and GA pathways highlighted the different biological stage between bulb conditions at T_3_: the uninfected bulbs were ready to release dormancy, whereas the OYVD-infected bulbs were not, due to the dormancy alteration caused by the virus. Genes related to brassinosteroids (BR), included in the *AcWRKY32* co-expression network, were mainly down-regulated in uninfected samples, except *SERK1* which showed a more than 5-fold up-regulation. The role of BR in plant growth-promoting is widely recognized [80,81], while a putative role of *SERK1* in bud dormancy release as well as response to stress was underlined in tree peony [82,83]. This gene is highly expressed under auxin or other plant growth regulators stimuli, demonstrating its involvement in plant development [84]. *SERK1* transcript abundance in the uninfected bulbs during dormancy suggested its involvement in meristem activities and tissue development after bulb dormancy release, as previously reported in other plants [83]. Altogether, the pathways identified in onion dormancy modulation, such as IAA and GA, as well as several key gene regulators up-regulated in our study at T_3_ (WRKYs, MYBs, *ABI5*, and *GA3ox1*), were reported to induce bulb dormancy release in *Lilium davidii* throughout an integrated transcriptomics and metabolomics analysis [85].

Finally, the results allowed us to develop a model for the potential interaction of genes included in the *AcWRKY32* co-expression network (Figure 6). This model provides a framework of the complex mechanism that regulates the breakage of dormancy in onion bulbs, triggered by a WRKY-TF member that drives cell wall modification and division through gibberellin and auxin homeostasis.

## 5. Conclusions

In summary, our findings shed light on the significant effect of the OYVD-infection on onion dormancy. ABA levels, and transcriptomic profiles provide evidences that virus-infected onion bulbs do not enter into a proper dormant state. Our results suggest that a significant up-regulation of *AcWRKY* family members is potentially required for dormancy breakage in onion. A complex network involving several key genes, mainly plant growth regulators, triggered by *Ac**WRKY* members suggest their pivotal contribution to dormancy release via cell wall modification and expansion through gibberellins and auxins homeostasis. In particular, *AcWRKY32* is co-expressed with several genes involved in the transcriptomic reprogramming during dormancy release in buds, seeds, and bulbs of other plants, supporting our conclusion. *AcWRKY32* and mainly its co-expressed genes might be considered the targets for controlling the early sprouting in onion, laying the foundations for novel breeding programs to improve shelf life and reduce postharvest losses due to this phenomenon. This study extended our understanding of the complex mechanisms involved in dormancy regulation and release in onion. Candidate genes identified here could be useful for further functional analyses of the *WRKY* gene family, hormone signaling, and regulation.

## Figures and Tables

**Figure 1 cells-11-01100-f001:**
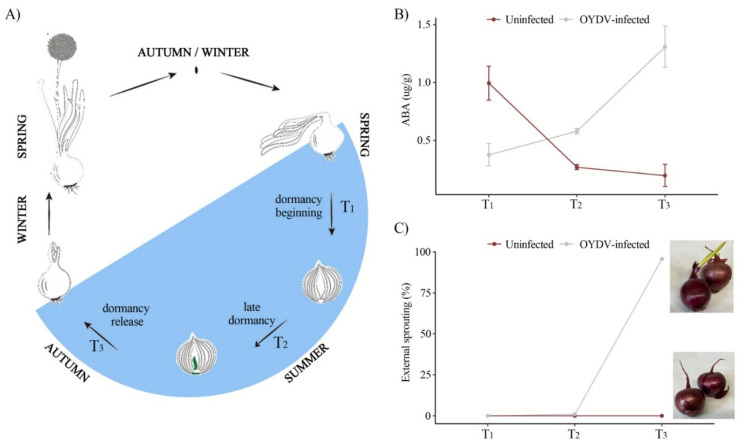
(**A**) Onion life cycle. The stages between the beginning of dormancy (T_1_), the late dormancy (T_2_), and the dormancy release (T_3_) are highlighted (in light blue); (**B**) ABA concentration (µg/g) evaluated through HPLC/MS; and (**C**) sprouting (%) in uninfected and OYDV-infected plants at the three time points investigated.

**Figure 2 cells-11-01100-f002:**
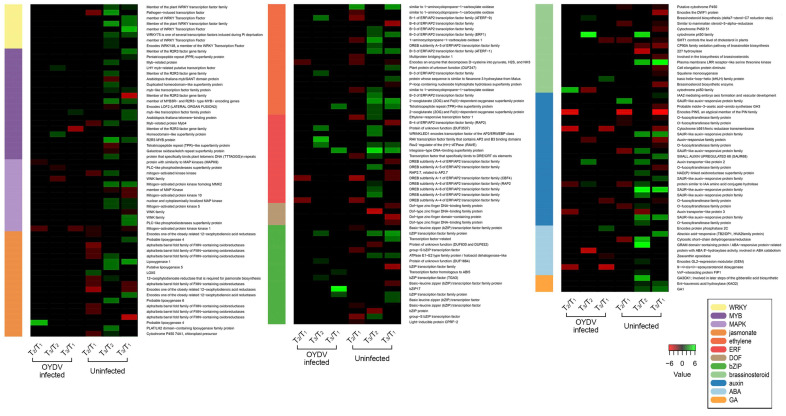
Heatmap developed using DEGs isolated in uninfected bulbs in the three comparisons T_2_/T_1_, T_3_/T_2_, and T_3_/T_1_. Log2 FC for uninfected and OYDV-infected conditions is displayed. DEGs for each class were listed in Appendix A.

**Figure 3 cells-11-01100-f003:**
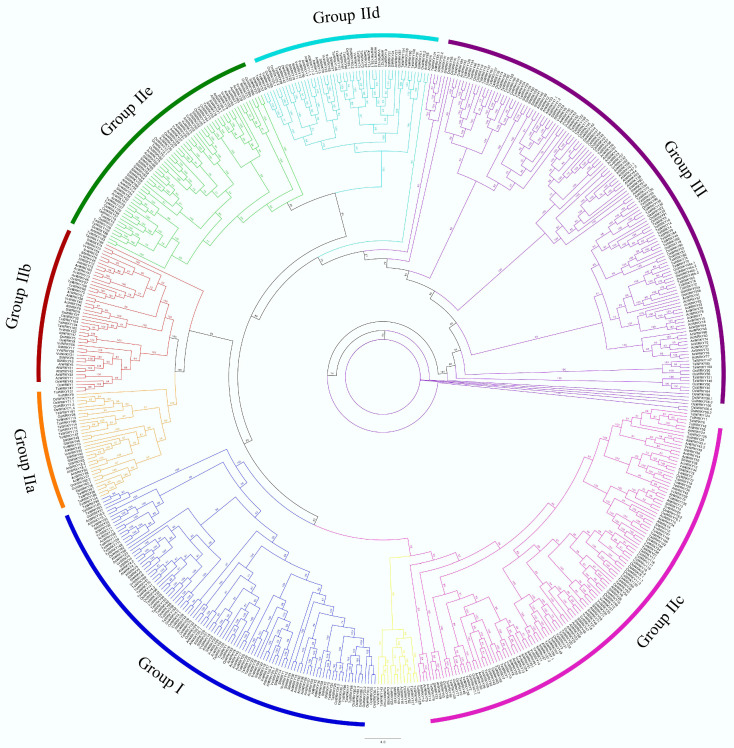
Neighbor-joining tree based on an alignment of 587 full-length genes encoding WRKY proteins, belonging to six plant species: *Allium cepa* (*AcWRKY*), *Arabidopsis thaliana* (*AtWRKY*), *Oryza sativa* (*OsWRKY*), *Vitis vinifera* (*VvWRKY*), *Triticum aestivum* (*TaWRKY*), and *Solanum lycopersicum* (*SlWRKY*) (for detail see Appendix A). Branch lines of subtrees are colored indicating different WRKY groups and subgroups (I, IIa, IIb, IIc, IId, IIe, and III).

**Figure 4 cells-11-01100-f004:**
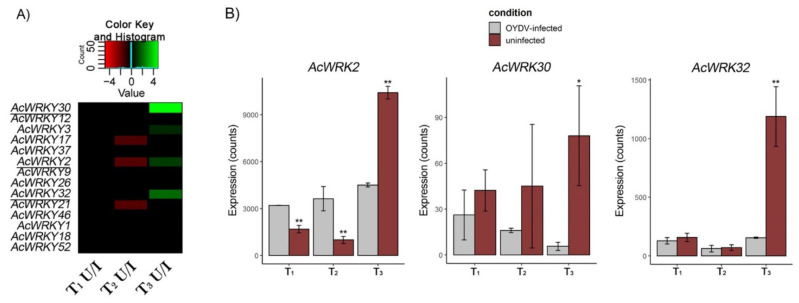
Expression levels of *AcWRKY* genes isolated in onion bulb at the beginning of dormancy (T_1_), late dormancy (T_2_), and the dormancy release (T_3_). (**A**) Heatmap developed using log2 FC values (by Deseq2) of *AcWRKYs* for uninfected (U)/OYDV-infected (I) bulbs compared at each time; *AcWRKYs* showing significant differences by RNASeq (*p* < 0.05; log2 FC > 1) and qPCR between the two conditions during dormancy release (T_3_) were underlined. (**B**) Normalized RNA-Seq counts of *AcWRKY2, 30*, and *32* at each sampling time, detected in uninfected (red) and OYDV-infected (gray) samples. ** *p* < 0.05, * *p* < 0.1 (between infected and uninfected bulbs).

**Figure 5 cells-11-01100-f005:**
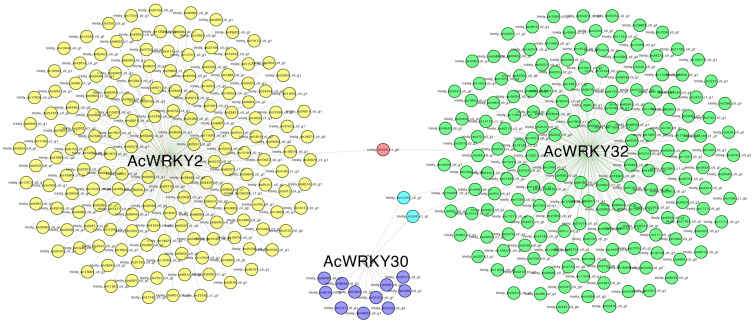
Co-expression network analysis of 3 *AcWRKYs* (*AcWRKY2, 30, 32*) up-regulated during dormancy release (T_3_) in the uninfected/OYDV infected bulb comparison. The network was obtained using the CoExpNetViz tool and visualized with Cytoscape. Green and red lines denote positive correlation and negative correlation, respectively. Details of genes involved in co-expression network analyses are provided in Appendix A.

**Figure 6 cells-11-01100-f006:**
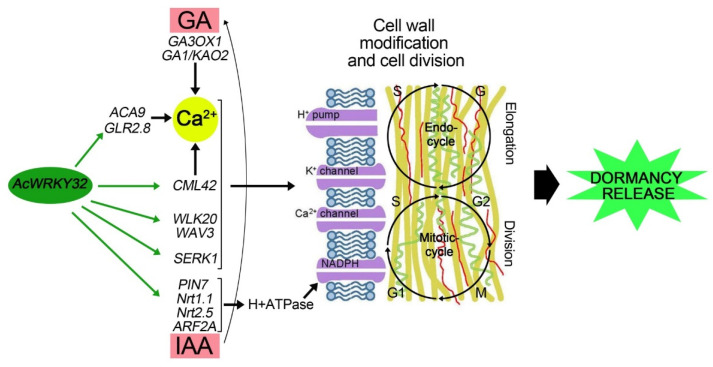
An overview of the proposed model explaining *AcWRKY32* and its co-expression gene network role in the onion bulb dormancy release. The model underlines the triggering role of the TF for driving cell wall modification and division through gibberellin and auxin homeostasis, two key processes involved in the dormancy release. The green arrows indicate the triggering action of *AcWRKY32* to its co-expressed genes. The black arrows underline the pathway steps determining the dormancy release previously described. The pink rectangles highlight the two crucial hormones (IAA and GAs) related to dormancy breakage, while the yellow circle underlines the key molecule (calcium) involved in the cell wall modification (elongation and expansion) and cell division. Differentially expressed genes putatively involved in dormancy breakage and interacting with *AcWRKY32* are reported in black.

**Table 1 cells-11-01100-t001:** List of WRKY genes per species used in the phylogenetic analysis: *Allium cepa* (*AcWRKY*), *Arabidopsis thaliana* (*AtWRKY*), *Oryza sativa* (*OsWRKY*), *Vitis vinifera* (*VvWRKY*), *Triticum aestivum* (*TaWRKY*), and *Solanum lycopersicum* (*SlWRKY*). Data downloaded from Plant TFDB [24].

Species	Group I	Group IIa	Group IIb	Group IIc	Group IId	Group IIe	Group III	Unclassified	Total
AcWRKY	16	2	8	9	2	11	25	1	74
AtWRKY	14	4	7	19	9	9	16	3	81
OsWRKY	24	8	5	18	10	14	40	2	121
VvWRKY	16	3	7	13	7	5	6	2	59
TaWRKY	45	12	7	34	17	10	40	6	171
SlWRKY	15	5	7	16	5	21	11	1	81
Total	130	34	41	109	50	69	138	15	587

## Data Availability

The data presented in this study are available in the text or Appendix A here.

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
