# Peer review of "WRKY Gene Family Drives Dormancy Release in Onion Bulbs"

_cells, 2022, doi:10.3390/cells11071100_

Round 1

Reviewer 1 Report

The manuscript “WRKY gene family drives dormancy release in onion bulbs” presented results about molecular analysis of dormancy release of onion bulbs cv. ‘Rossa di Tropea Onion Yellow Dwarf Virus infected and uninfected plants. The results provide for the first time, by using de novo assembeled transcriptome data, a comprehensive overview of the WRKY family members in onion. According to applied analysis it is suggested that a significant up-regulation of WRKY family members is potentially required for dormancy breakage in onion.

The manuscript falls in a scope of journal and can be accepted for publication after minor corrections:

Line 648, 649: Why germination and seedlins establishment is in blue and underline

Line 692: there is extra space between of and the.

Author Response

The manuscript “WRKY gene family drives dormancy release in onion bulbs” presented results about molecular analysis of dormancy release of onion bulbs cv. ‘Rossa di Tropea Onion Yellow Dwarf Virus infected and uninfected plants. The results provide for the first time, by using de novo assembled transcriptome data, a comprehensive overview of the WRKY family members in onion. According to applied analysis it is suggested that a significant up-regulation of WRKY family members is potentially required for dormancy breakage in onion.

Authors answer: thanks so much for appreciating the manuscript by the reviewer. All revisions in the text have been highlighted in red. 

The manuscript falls in a scope of journal and can be accepted for publication after minor corrections:

Line 648, 649: Why germination and seedlings establishment is in blue and underline

Authors answer: thank you, it has been modified, it was certainly a typo

Line 692: there is extra space between of and the.

Authors answer: thank you, it has been deleted the extra space.

Reviewer 2 Report

Authors have showed the role of WRKY gene family in regulation of dormancy in onion bulbs in manuscript entitled “WRKY gene family drives dormancy release in onion bulbs”.

The work has presented some novel aspects of WRKY gene family in releasing the dormancy in onion bulbs. Authors also first time claimed, a detailed molecular analysis of the dormancy process, a description of the WRKY-TF family in onion, and providing a better understanding of the putative role played by AcWRKY32 in the bulb dormancy release. Beyond the novelty of the work, manuscript is also found with major concerns. Moreover, English language needs moderate improvements. Punctuations and spacing also need to be checked carefully throughout the manuscript.

Abstract:

The main feature of the research study and how it can impact future research should be highlighted in the abstract.

Word selection in keywords should be different from title of the manuscript. Please consider adding new words by replacing existing words. Also, remove abbreviations from the keywords.

Please present abstract in one para as per editorial requirement of journal.

Introduction:

Any previous studies (recent one) on similar area must be highlighted in the introduction section. Production’s aspects of onion will add value to the introduction. While reviewing I found that literature cited is quite older. Only one reference from 2021 and very few from 2020 and 2019.  

Line no. 32: Abbreviations need to be defined at first. This comment applies throughout the manuscript.

Material and methodology:

Material section need to be added. Various material used in the study are not mentioned.

In materials, make of many chemicals are not mentioned; consider adding it so to help others to replicate the experiment.

Authors must add a flow diagram showing sequence of experimentation done in the current investigation and what are various test/analysis done during the scheme of experimentation. This flow diagram will definitely improve the readability of the manuscript.

Result and Discussion:

Results can be well followed but I am not satisfied discussion given by the authors to justify their findings. It is suggested to improve the discussion of the manuscript with recent literature. I wish to see a revised version of the manuscript with a good discussion throughout the result and discussion section.

The figures such as fig. 2, fig. 3 and 5 are showing important information related to results of the study but they are not readable due to very small size. Authors need to find a way to improve the figures so that it can help to better understand the results.

Conclusion:

Future perspective needs to be added in the manuscript. It will help other researcher to work further in this area.

Author Response

Authors have showed the role of WRKY gene family in regulation of dormancy in onion bulbs in manuscript entitled “WRKY gene family drives dormancy release in onion bulbs”.

The work has presented some novel aspects of WRKY gene family in releasing the dormancy in onion bulbs. Authors also first time claimed, a detailed molecular analysis of the dormancy process, a description of the WRKY-TF family in onion, and providing a better understanding of the putative role played by AcWRKY32 in the bulb dormancy release. Beyond the novelty of the work, manuscript is also found with major concerns. Moreover, English language needs moderate improvements. Punctuations and spacing also need to be checked carefully throughout the manuscript.

Authors answer: thanks so much for appreciating the manuscript by the reviewer, we are also confident to have addressed adequately the major concerns underlined by the reviewer starting from the English style improvement as well as the punctuations and spaces overall the manuscript. All revisions in the text have been highlighted in red.

Abstract:

The main feature of the research study and how it can impact future research should be highlighted in the abstract.

Authors answer: thank you for the suggestion. We have addressed the requested in the abstract including a new sentence that highlight the future perspective of research.  

Word selection in keywords should be different from title of the manuscript. Please consider adding new words by replacing existing words. Also, remove abbreviations from the keywords.

Authors answer: thanks so much for this suggestion; the keywords have been modified as suggested.  

Please present abstract in one para as per editorial requirement of journal.

Authors answer: thanks so much for this suggestion; it has been done.  

Introduction:

Any previous studies (recent one) on similar area must be highlighted in the introduction section. Production’s aspects of onion will add value to the introduction. While reviewing I found that literature cited is quite older. Only one reference from 2021 and very few from 2020 and 2019.

Authors answer: thank you for this suggestion; albeit there are not many recent studies exactly in the focus of our manuscript, we produced the efforts requested to include some others recent references. Interestingly, many recent works related to dormancy on bulbous species and/or WRKY have confirmed the evidences already highlighted in the past years as well as in our manuscript.

Line no. 32: Abbreviations need to be defined at first. This comment applies throughout the manuscript.

Authors answer: thank you for the suggestion; it has been done.

Material and methodology:

Material section need to be added. Various material used in the study are not mentioned.

Authors answer: thank you for the suggestion; as required we added some information in the new version of the manuscript. All refuses and typos were modified and all the unclear points in the materials and methods were improved.

In materials, make of many chemicals are not mentioned; consider adding it so to help others to replicate the experiment.

Authors answer: thank you for the suggestion; we added the makes of the chemicals utilized for the experiments where it was needed.

Authors must add a flow diagram showing sequence of experimentation done in the current investigation and what are various test/analysis done during the scheme of experimentation. This flow diagram will definitely improve the readability of the manuscript.

Authors answer: thank you for the suggestion; we defined a flow chart of our experimental setup (Figure S7 in the new version submitted).

Result and Discussion:

Results can be well followed but I am not satisfied discussion given by the authors to justify their findings. It is suggested to improve the discussion of the manuscript with recent literature. I wish to see a revised version of the manuscript with a good discussion throughout the result and discussion section.

Authors answer: thank you for the suggestion; we have sustained large efforts in improving the discussion of the manuscript with some more recent citations.

The figures such as fig. 2, fig. 3 and 5 are showing important information related to results of the study but they are not readable due to very small size. Authors need to find a way to improve the figures so that it can help to better understand the results.

Authors answer: thank you for the suggestion; the figures listed above are in high definition, and due to the margins to be included in the text it is not possible to improve the size. For this reason all the information (gene names, annotation, expression values etc.) were reported (as in the original submission) in the supplementary tables. However, we included some additional information in the caption for some of them, for helping the readers to better understand our results.

Conclusion:

Future perspective needs to be added in the manuscript. It will help other researcher to work further in this area.

Authors answer: thank you for this important suggestion; we have added some important sentences for guiding the readers to the future perspective.

Reviewer 3 Report

A very interesting paper that presents some foundational results to our understanding of dormancy breaking in onion. The study is very thorough. 

Could the authors speculate on the results if they had used a long-day onion cultivar as opposed to a short-day onion cultivar since the dormancy period would occur at a different time of year? Also since Onion Yellow Dwarf Virus altered the dormancy response in this cultivar would other virus such as Iris yellow spot virus also cause a similar response? The manuscript would be strengthened if a second cultivar had been examined so that a similar response could have been observed in cultivars with different genetic backgrounds. I recognize that this additional cultivar would have expanded the project exponentially. 

With the frequency that some supplemental tables and figures are referenced in the text, it seems more appropriate to put them in the manuscript.

Please see the attached file for some grammatical corrections.

Author Response

A very interesting paper that presents some foundational results to our understanding of dormancy breaking in onion. The study is very thorough.

Authors answer: thanks so much for appreciating the manuscript by the reviewer. All revisions in the text have been highlighted in red.

Could the authors speculate on the results if they had used a long-day onion cultivar as opposed to a short-day onion cultivar since the dormancy period would occur at a different time of year? Also since Onion Yellow Dwarf Virus altered the dormancy response in this cultivar would other virus such as Iris yellow spot virus also cause a similar response? The manuscript would be strengthened if a second cultivar had been examined so that a similar response could have been observed in cultivars with different genetic backgrounds. I recognize that this additional cultivar would have expanded the project exponentially.

Authors answer: thanks so much for your comment; based on your last consideration it is really difficult to speculate on what the reviewer is interested without the second genotypes included in the experimental setup. The cv ‘Rossa di Tropea’ and OYDV have been chosen due to the importance and disease incidence in Italy. However, due to the lack of knowledge on this topic, this is the first evidence about the mechanism involved in the dormancy release, and the system here verified can be utilized with different genotypes and virus in further studies.

With the frequency that some supplemental tables and figures are referenced in the text, it seems more appropriate to put them in the manuscript.

Authors answer: thanks so much for this comment; there are many supplementary materials that are frequently referenced in the text, but with secondary information. The figures and tables included in the main text show the key information for manuscript understanding that we retain necessary and sufficient, albeit fewer cited in the text. In addition, the tables cited many times are of big size and they cannot be included in the main text (e.g. Table S5 and S6).

Please see the attached file for some grammatical corrections.

Authors answer: thanks so much, all the grammatical corrections have been made as suggested.

Round 2

Reviewer 2 Report

Authors have improved the quality of the manuscript. May be accepted for publication.